# Changes on Techno-Functional, Thermal, Rheological, and Microstructural Properties of Tef Flours Induced by Microwave Radiation—Development of New Improved Gluten-Free Ingredients

**DOI:** 10.3390/foods12061345

**Published:** 2023-03-22

**Authors:** Caleb S. Calix-Rivera, Marina Villanueva, Grazielle Náthia-Neves, Felicidad Ronda

**Affiliations:** 1Department of Agriculture and Forestry Engineering, Food Technology, College of Agricultural and Forestry Engineering, University of Valladolid, 34004 Palencia, Spain; 2Department of Agroindustrial Engineering, Pacific Littoral Regional University Center, National Autonomous University of Honduras (UNAH), Choluteca 51101, Honduras

**Keywords:** tef, gluten-free flours, microwave radiation, flours modification, physico-chemical properties

## Abstract

Tef [*Eragrostis tef* (Zucc.) Trotter] flour is a gluten-free cereal rich in fiber, minerals, vitamins, and antioxidants, which offers a promising alternative for new food development. This study investigated the effect of microwave radiation (MW) on the techno-functional, thermal, rheological and microstructural properties of tef flours. White and brown tef grains were milled and microwaved at different moisture contents (MC) (15%, 20% and 25%) for a total irradiation time of 480 s. The morphological structure of tef flours was affected by MW treatment, and its particle size and hydration properties increased after the treatment. Lower peak, breakdown, and setback viscosities, up to 45%, 96%, and 67% below those of the control (untreated) samples, and higher pasting temperature, up to 8 °C in the 25% MC samples, were observed. From FTIR analysis a disruption of short-range molecular order was concluded, while DSC confirmed an increased stability of starch crystallites. Rheological analysis of the gels made from the treated samples revealed that MW had a structuring and stabilizing effect on all samples, leading to higher viscoelastic moduli, G′ and G″, and the maximum stress the gels withstood before breaking their structure, τ_max_. The MC of the flours during the MWT drove the modification of the techno-functional properties of the tef flours and the gel rheological and thermal characteristics. These results suggest that MW-treated tef flours are potential ingredients for improving the technological, nutritional and sensory quality of food products.

## 1. Introduction

Currently, the global prevalence of celiac disease, an autoimmune disease related to the consumption of gluten protein, is approximately 1–2% [1]. In addition to celiac individuals, some people avoid consuming gluten due to a personal preference or other medical condition. All these aspects have driven the demand for gluten-free (GF) products. According to Statista [2], the global GF products market size was valued at USD 6.7 billion in 2022 and is expected to expand considerably between 2023 and 2032, from USD 7.2 billion to USD 14 billion.

Despite the existing advances to improve the quality of GF foods, which include the use of starches, gums, hydrocolloids and others [3], the formulation of GF products still represents a great challenge for the cereal technologist. This is because many of the products available have poor nutritional value, a bad feeling of mouth or taste, and, not the least, they are quite expensive compared with traditional baked goods [4]. Therefore, there is a need to discover alternative technologies for improving the techno-functional characteristics of GF-based foods as well as popularizing little-known GF cereals sources.

Recently, tef [*Eragrostis tef* (Zucc). Trotter], an ancient GF crop traditionally cultivated in Eritrea and Ethiopia, has been gaining popularity in the global market due to its high nutritional value and its technological versatility for producing different products such as injera, opaque beer, spirits, porridge, and atmit intended for people who have a gluten-restricted diet [5]. Nutritionally, this cereal has a low glycemic index and is richer in protein, essential amino acids, minerals, and soluble fiber than other cereals such as sorghum, maize, pearl millet, barley, and rice [6,7]. In addition, recent studies have also shown that tef has better functional properties (foaming and water absorption capacity), a longer shelf life and slower aging of its baked goods than wheat, sorghum, rice, barley, quinoa, oat, and maize [8,9,10]. For all these reasons, tef cultivation has expanded to other countries outside Africa, which include the United States of America, Canada, Spain, Australia, and Switzerland [9]. However, the existing studies about the impact of technological processing on this cereal are still limited and have yet to be adequately studied. Considered the smallest cereal grain in the world (1 mm in length), tef is eaten and processed as a whole grain. In general, the use of whole and GF grains for bread-making represents a challenge for food scientists, as they may negatively impact the sensory quality and technological aspects of bread, such as the volume and firmness of crumbs [11]. Over the last decades, several studies have been published on innovative methods and technologies successfully applied to improve the quality of GF-based foods.

Among the alternatives, microwave (MW) processing represents a rapid, green, efficient, and reliable technique to physically modify flours and develop new bakery products with improved functionality [12,13]. For example, Solaesa, et al. [14] studied the physical modification of rice flours by MW treatment (MWT) and found significant changes in the chemical (increase in amylose content), rheological (more stable gels), and thermal properties (increasing pasting temperature) of the treated samples. Villanueva, et al. [13] also reported improvements in the dough (viscoelastic behavior) and bread (softer crumb due to higher specific volume, and delayed staling) prepared from microwaved rice flour. Moreover, the MWT has enhanced some functional properties such as water absorption index, water solubility index, swelling power, water absorption capacity, and emulsion stability of buckwheat grains treated at 13% moisture content (MC) [15]. All these changes/improvements promoted by MWT can be associated with the mechanism of heat and mass transfer or due to the interactions between MW radiation and the individual polar (water) and ionic molecules (mainly mineral salts) [16]. Unlike conventional heat transfer mechanisms (conduction and convection), the heat transfer by MW occurs by convection and radiation, which provide rapid, homogeneous, and volumetric heating inside the treated sample [17].

Although significant research has been carried out to understand the changes pro-moted by MW on starches and flours, it is still necessary to optimize/explore process conditions in different sources of flours/starches to diversify the use of this technology in the production of GF foods. According to the literature, the available water content and the heating rate can significantly affect the physicochemical properties of cereal starches and, consequently, influence the quality of bakery products [13,18]. To the best of our knowledge, there has not been any study in the scientific literature regarding the impact of MWT on moistened tef flours. There is a need to better understand the full impacts of MWT on the tef matrix as an important step in turning this little-known GF cereal into a valuable ingredient to be included in the formulation of GF products.

Thus, the purpose of this study was to evaluate the impact of the MW-assisted thermal treatment on the techno-functional, structural, thermal, and rheological properties of tef flours as a function of its moisture content (MC) and ecotype used. The results obtained from this work will support cereal technologists aiming to improve the functional proper-ties of tef-derived products.

## 2. Materials and Methods

### 2.1. Tef Flours

Tef [*Eragrostis tef* (Zucc)] grains of two Spanish ecotypes, white and brown, kindly provided by CYLTEF (Zamora, Spain), were used for this study. Both tef grains were milled using a Perten Instrument mill (LM 3100, Hägersten, Sweden) with a 500 μm aperture sieve. The milled flours were stored at 4 °C until MWT. The proximate composition of white and brown tef flours were 12.92% and 12.71% moisture; 2.18% and 2.61% lipids; 9.38% and 9.94% proteins; and 6.7% and 7.03% dietetic fiber, respectively. Moisture content was measured by the official AACC Method 44–19 [19]. AOAC methods were employed for the determination of lipid content (n° 923.05–1923) [20], protein content (N × 5.7) (n° 960.52–1961) [21] and dietary fiber (n° 991.43–1994) [22].

### 2.2. Microwave Treatment (MWT)

First, both tef flours were moistened with distilled water to reach 15%, 20%, and 25% ± 0.5% of MC. In the treatments, 50 g of each tef flour was exposed to MW radiation (900 W Sharp MW oven R342 (Sakai, Japan)) for 480 s in cycles (10 s radiation/50 s of rest) for a total time of 48 min for each treatment, in a hermetic container in continuous rotation using an external device set at 70 rpm to ensure uniform energy and temperature distribution during treatment. The maximum temperatures reached during treatments, determined using TESTO Testoterm^®^ temperature strips (Madrid, Spain), were: 117 ± 5 °C (15% MC), 122 ± 5 °C (20% MC), and 138 ± 5 °C (25% MC). Once treated, the flours were dried at 35 °C until reaching their natural MC (~12%) and sieved to <500 μm for further analysis. Samples were named WTF-15, WTF-20, WTF-25 and BTF-15, BTF-20, BTF-25 for white (WTF) and brown (BTF) tef samples, respectively. Untreated white and brown tef flours were used as controls. Each treatment was performed in quadruplicate.

### 2.3. Scanning Electron Microscopy (SEM)

SEM was used to analyze the surface microstructure of tef flours before and after the MWT. SEM analysis was performed on a Quanta 200FEG scanning electron microscope (FEI, Hillsboro, OR, USA) equipped with an X-ray detector. The samples were prepared by mounting small pieces of film onto aluminum stubs using conductive carbon tape and sputter-coated with a 5 nm layer of gold using an SCD–05 Leica Microsystems (Wetzlar, Germany). Visualizations were performed at an accelerating voltage of 7 keV in low vacuum mode using a secondary electron detector at 100×, 500×, 1500×, and 3000× magnifications. Representative micrographs were selected to illustrate the microstructure modifications.

### 2.4. Particle Size Distribution

A Mastersizer 3000 laser diffraction particle size analyzer (Malvern, UK) was used to determine the samples’ particle size distribution. Results were expressed as median diameter (D_50_) and dispersion ((D_90_ − D_10_)/D_50_) as described in Abebe et al. [10]. All flours were analyzed in triplicate.

### 2.5. Damaged Starch and Amylose Content

The damaged starch content of treated and untreated tef flours was determined following the AACC Official Method 76–31.01 [23] with a Megazyme KSDAM starch damage Kit (Bray, Ireland). The damaged starch was expressed as g/100 g of tef flour dry matter (d.m). The amylose content (AC) was determined by the lectin concanavalin A (Con A) method [24] with the Megazyme K–AMYL assay kit (Bray, Ireland) and was expressed as g/100 g of starch. In both methods, the absorbance was read at 510 nm, and the samples were analyzed in triplicate.

### 2.6. Hydration Properties

Water absorption capacity (WAC), water absorption index (WAI), water solubility index (WSI), and swelling power (SP) of untreated and microwaved tef flours were determined following the protocol described by Abebe et al. [10] with slight modifications. For WAC, two grams of flour dry matter were mixed with 20 mL of distilled water in 50 mL centrifuge tubes. The dispersions were occasionally vortexed while they were held at room temperature for 30 min, followed by centrifugation for 25 min at 3000× *g* (Thermo Fisher Scientific, Waltham, MA, USA). The supernatant was removed, and the remainder was weighed; WAC was expressed as g H_2_O/g flour (d.m). To determine the WAI, WSI, and SP, the mixtures were boiled for 15 min and cooled down to room temperature before being centrifuged at 3000× *g* for 10 min. The supernatant was poured in a previously weighed evaporating capsule to determine its solid content, and the sediment was weighed. The weight of the soluble solids was determined by evaporating the water from the supernatant overnight at 110 °C; WAI was expressed as g sediment/g flour (d.m), WSI was expressed as g soluble solids/100 g flour (d.m), and SP was expressed as g sediments/g insoluble solids in the flour (d.m). All hydration properties were measured in triplicate.

### 2.7. Pasting Properties

The pasting properties of the studied tef flours were determined following the official method AACC 76–21.02 STD2 [25] using a Kinexus Pro+ rheometer (Malvern, UK) equipped with a starch pasting cell geometry. Each flour (3.5 g, 14% moisture basis) was mixed with 25 g of distilled water before being loaded into the starch cell. A temperature of 50 °C was applied for 1 min, followed by heating to 95 °C at a rate of 6 °C/min, holding at 95 °C for 5 min, cooling to 50 °C at a rate of 6 °C/min, and holding at 50 °C for 2 min. The paddle speed was set at 160 rpm. Pasting temperature (PT), peak viscosity (PV), trough viscosity (TV), final viscosity (FV), breakdown (BV = PV − TV), and setback (SV = FV − TV) parameters were recorded. Each flour was analyzed in duplicate.

### 2.8. Thermal Properties

Thermal properties of the untreated and treated tef flours was determined by a differential scanning calorimeter (DSC3, STARe-System, Mettler-Toledo, Greifensee, Switzerland). For that, flours (~6 mg) with an excess of water (70%) were weighed in a 40 μL aluminum pan and heated from 0 to 115 °C at 5 °C/min using an empty sealed pan as reference. The onset (T_O_), peak (T_P_), and endset (T_E_) temperatures and the enthalpy of gelatinization and amylose-lipid dissociation (ΔH) (J/g flour d.m) were recorded. Once heated (gelatinized), the flour samples were kept under refrigeration (4 ± 2 °C) in the pans and after 7 days were scanned a second time following the same procedure described above to evaluate their retrogradation. Each measurement was performed in duplicate.

### 2.9. Fourier Transform Infrared Spectroscopy (FTIR)

FTIR spectra of the studied tef flours were recorded by a FT-IR Nicolet iS50 spectrophotometer (Thermo Fisher Scientific, USA) equipped with a crystal diamond attenuated total reflectance (ATR) sampling accessory. The samples’ humidity was set at 15% MC using a saturated humidity Memmert ICP260 incubator (Schwabach, Germany). Measurements were performed in the range of 400–4000 cm^−1^ with a resolution of 4 cm^−1^ and an accumulation of 64 scans. The short-range molecular ordered structure of starch (ratios of absorbance 1047/1022 cm^−1^ and 1022/995 cm^−1^) and amide I bands (1700–1600 cm^−1^) were analyzed by Fourier self-deconvolution using OMNIC 9 software (Thermo Fisher Scientific Inc. USA). Two points straight line baseline correction followed by 2nd order derivative of spectra for peak finding and final Gaussian peak fitting at those peak positions were performed in Origin 2019b (OriginLab Corporation, Northampton, MA, USA) to calculate percent contribution by secondary structure components. Peaks were classified as: β-sheet (high frequency) (1700–1690 cm^−1^), β-turns (1690–1665 cm^−1^), random coil and α-helix (1665–1640 cm^−1^), and β-sheet (low frequency) (1640–1615 cm^−1^) [26]. Measurements were performed in triplicate.

### 2.10. Nuclear Magnetic Resonance Spectroscopy (NMR)

^1^H NMR analyses were performed using a 500 MHz NMR spectrometer (Agilent Instruments, USA) equipped with OneNMR probe at 70 °C, 45° pulse width, spectral width of 8012.8 Hz, a total of 400 transients, acquisition time of 2.004 s, and a relaxation delay of 5 s. Samples were prepared according to the procedure described by Acevedo, et al. [27]. MestReNova software v.12 (Mestrelab Research Co., Santiago de Compostela, Spain) was used to analyze spectra. Degree of branching (DB) in the studied samples was determined from the results of ^1^H NMR following the Equation (1). All measurements were performed in duplicate.
(1)$$DB\left(\%\right)=\frac{I_{\alpha -(1,6)}}{I_{\alpha -(1,4)}+I_{\alpha -(1,6)}}\times 100$$
where I_α−(1,6)_ is the area under the curve of the peak corresponding to α−(1,6)–glucosidic bonds at ~4.80 ppm, and I_α−(1,4)_ is the area under the curve of the peak corresponding to α−(1,4)–glucosidic bonds at ~5.12 ppm [27].

### 2.11. Rheological Properties

Rheological properties of the tef flour gels were determined with a Kinexus Pro+ rheometer (Malvern Instruments Ltd., Malvern, UK), using a 40 mm diameter serrated parallel plate geometry at 1 mm gap. The gels of each flour were analyzed 10 min after processing (following the procedure described in Section 2.7). The gels were placed between plates and allowed to rest for 5 min. The temperature was stabilized at 25 °C using a Peltier plate controller. Strain sweeps were carried out from 0.1 to 1000% strain at a constant frequency of 1 Hz. From them, the maximum stress (τ_max_) beyond which the dough structure was broken (corresponding to the linear viscoelastic zone, LVR) and the stress at the crossover point (G′ = G″) were stablished. Frequency sweeps were made from 10 to 1 Hz at 1% strain (in the LVR). The data obtained from frequency sweep were fitted to the power law model [28]. The fittings coefficients, G_1_′, G_1_″, and (tan δ)_1_, which represent the elastic and viscous moduli and the loss tangent at 1 Hz, respectively, as well as the exponents of the potential equations, *a*, *b*, and *c*, which represent the degree dependence of these moduli and the loss tangent with oscillation frequency, were obtained. Tests were carried out in duplicate.

### 2.12. Statistical Analysis

Statistical analysis was performed with analysis of variance (ANOVA) of the obtained results by the Least Significant Difference (LSD) test at *p*-value < 0.05 using Statgraphics Centurion XVIII software (Bitstream, Cambridge, MN, USA).

## 3. Results and Discussion

### 3.1. Morphology and Particle Size Distribution

SEM images of untreated and treated tef flours at 25% MC are shown in Figure 1. Samples with higher moisture content were presented, since these showed the highest changes in the other properties analyzed in relation to the untreated samples. The untreated flours (Figure 1A,C) showed the typical tef flour morphology, which consists of simpler polygonal starch granules (2–6 μm in diameter) with a smooth surface, and packed by globular protein and lipids [10,29,30]. The samples submitted to the MW process at 25% MC showed a significant difference in size and shape concerning the untreated ones. After treatment, the flours presented more agglutinated starch granules with rough and slightly swollen structures (Figure 1B,D). Zavareze and Dias [31] also reported that rice starch submitted to heat–moisture treatment (HMT) at 25% MC showed granules that were more aggregated and had a more irregular surface than untreated samples. The observed increase in the swelling power (SP) of the starch granules in the treated flours (Figure 1B,D) might be due to molecular reorganizations that occur during treatments with high MC for a prolonged period at a temperature above that of glass transition [32]. An increase in the SP was also reported by Deka and Sit [33] for taro starch after HMT. The 500× and 1500× magnifications from the treated samples [Figure 1B2,B3,D2 and D3] revealed larger particles and no rupture of the starch granules. Additionally, the 3000x magnifications indicated agglomeration of starch granules, despite that thermal properties discarded its partial gelatinization (see Section 3.5) [34].

The particle size distributions of the treated and untreated flours are presented in Table 1 and Appendix A. The results of the mean diameter (D_50_) and size distribution ((D_90_ − D_10_)/D_50_) confirm that MWT promoted an increase in particle size as observed by SEM images (Figure 1). The untreated tef samples (WTF and BTF) presented similar D_50_ values. A significant increase in the particle size and a significant decrease in the size dispersion were observed in both treated flours. The effect of MWT on the particle size was more prominent in the white tef (WTF) than in the brown tef (BTF). WTF treated at 15%, 20%, and 25% MC showed a 25%, 30%, and 45% increase in D_50_, respectively, compared to untreated samples, while the treated BTF samples only showed a 12–17% increase in this parameter over all the MC levels studied. This increase in the particle size observed after the MWT might be associated with starch granules agglomeration and protein denaturation that occur during heating processes [35]. According to the literature, heat treatments may favor the adhesion of denatured proteins to the surface of modified starch granules, leading to an increase in granule size [17,36]. Similar to D_50_, the size dispersions were also progressively affected by increasing MC in both treated flours, with the most significant changes observed in the WTF at 25% MC (29% reduction concerning the untreated WTF) and BTF-20 (16.7% reduction concerning the untreated BTF) samples. These reductions indicate that samples submitted to MW radiation presented more uniform particle sizes than the untreated flours.

### 3.2. Damaged Starch and Amylose Content (AC)

Damaged starch and amylose content (AC) of studied flours are shown in Table 1. Analysis of variance (*p* < 0.001) showed that the MC and the interaction between the tef ecotype and MC significantly affected the damaged starch content. The damaged starch content decreased with MW radiation, with the most significant changes observed for samples treated at 15% MC. At this treatment condition, the damaged starch from the WTF and BTF was reduced by 21% and 33%, respectively. These results are in agreement with Liu, et al. [37], who indicated that the damaged starch content from mung bean, potato, corn, and waxy corn starch was significantly reduced after HMT. These damaged starch reductions promoted by thermal processing could be explained by rearrangements of the amorphous region in damaged starch granules [37]. On the other hand, the AC from both tef flours increased after the MWT. For the WTF and BTF treated samples, the AC increased in the range 11–22% and 24–36%, respectively, as the MC increased. An increase in AAC from MW-treated taro starch was previously reported by Deka and Sit [33], who explained that the degradation of amylopectin led to many short-amylose chains. It can also be seen from Table 1 that no significant differences were observed for either damaged starch or AC measured between the two untreated tef flours, and that, similar to particle size analyses, the brown tef was more affected by the treatment than the white tef.

### 3.3. Hydration Properties

The values determined for WAC, WAI, WSI, and SP of treated and untreated tef flours are shown in Table 1 and Appendix A. The untreated samples showed similar hydration properties. The MC of tef flour and interaction between MC and tef ecotype significantly affected all hydration properties of both treated tef flours (*p* < 0.001). WAC values were higher for both treated samples and increased with increasing MC by more than 60% (at 25% MC) compared to the untreated flours. This is in accordance with SEM images (Figure 1), which showed that the MWT promoted the agglutination of the starch granules. Likewise, an increase in the WAC was observed in red bean and wheat flour after MWT [38]. This increase in WAC from treated samples may be related to the disruption of hydrogen bonds between the amorphous and crystalline starch regions during the MWT, which promotes amylose exposure and favors water binding [39]. Similar to WAC, the WAI, WSI, and SP properties increased in all treated samples with respect to untreated flours. All of these parameters showed a higher increase in samples wetted at lower moisture levels (15% and 20% MC), while samples treated at 25% MC level showed a lower increase (except for the WSI in the BTF). Similarly, previous reports indicated a significant increase in the WSI and SP values from chickpea flour and durum wheat semolina after MWT [40,41]. The changes observed in WAI and WSI properties can be attributed to macromolecular disorganization and starch degradation, as well as to the higher number of small amylopectin or amylose fragments leaking through the starch granules opened by the treatment [42,43]. The observed increases in SP have been related to modifications caused to the amorphous regions of the starch granules, as well as to changes in crystallinity during the hydrothermal treatment [13]. Thus, intense treatments may cause a structural rearrangement of the amylose and amylopectin molecules, as well as break intermolecular bonds, which contributes to a higher exposure of hydroxyl groups, and, as a consequence, allows a higher water uptake and retention in treated samples [33]. These findings can bring benefits to the formulation of bakery products. For instance, Villanueva, et al. [13] positively related the increase in WAC and SP properties with improvements in dough consistency and structure.

### 3.4. Pasting Properties

The pasting properties of treated and untreated tef flours are shown in Table 2 and Figure 2. The pasting behavior of the untreated tef flours agrees with that previously re-ported for other tef varieties [44]. A more significant effect was observed in flours treated with higher MC. This observation is consistent with those previously reported in HMT-treated starches [31]. Pasting temperatures (PT) increased significantly (*p* < 0.05) in all treated samples (except in WTF at 15% MC). As presented in Table 2, a positive effect of increasing MC of tef flours on the PT values was observed (increased by +4 °C and +6 °C in WTF and BTF treated at 25% MC, respectively). Higher PT indicates higher resistance of the starch to swelling and rupture. Such resistance can be attributed to the strengthening of intragranular bonding forces occurring during the treatment [36] as well as the changes in crystallinity that starch undergoes during MW treatment, and this is more pronounced with longer treatment duration [45]. Thus, the stronger these interactions, the more energy required to damage the starch structure and past formation [46]. The results obtained here are in agreement with other authors who have reported a positive effect on increasing the gelatinization temperature of wheat, corn, and other starches as a result of MW processing [47].

Opposite to PT, peak viscosity (PV), trough viscosity (TV), final viscosity (FV), breakdown viscosity (BV), and setback viscosity (SV) decreased in all samples submitted to the MWT, with the effect being more pronounced at higher MC. All these viscosity parameters were significantly affected by the interaction between MC and tef ecotype (*p* < 0.001). Similar observations on the decrease of these pasting properties was found by Villanueva, et al. [17] and Calix-Rivera, et al. [48] in microwaved rice flours. Changes in pasting properties may be the result of structural rearrangements and starch chain association during microwave heating [15]. Solaesa, et al. [14] and Sun, et al. [46] related the reduction of PV with an increment of inter- and intra-molecular hydrogen bonding due to association of starch chains during MWT. Moreover, Quin, et al. [49] related a relationship between the PV and the size of the granules. According to these authors, the PV of rice flour increased as the particle size of the granules decreased, which they attributed to the hydration susceptibility of smaller granules to gelatinization. In agreement with this report, in this present study, the increasing particle size with MC (Section 3.1) could contribute to the decrease in viscosity observed in the microwaved samples. Lower BV values reflect an increase in shear stability of microwave-heat-treated flour [46] Similarly, Abebe et al. [45] reported a significant decrease in the BV values from brown tef flour after large times of MWT. Lower SV values indicate lower amylose retrogradation, which may improve the application of these treated flours as thickening agents for products such as soups or sauces, due to its lower tendency for syneresis and more stability toward heating and cooking [15], The decrease in SV may be due to the generation of new interactions between amylose and amylose and/or amylopectin–amylose chains due to MWT, leading to a reduction in amylose leaching and thus a reduction in setback viscosity [17]. The lower FV observed in the treated samples may be due to the shortening of the amylopectin branch chains and decreased polymerization of amylose and amylopectin caused by the MWT [50]. Thus, from these results, it can be inferred that the changes in the pasting properties can be explained by the associations between chains in the amorphous region of the granule as well as by the changes in crystallinity during hydrothermal treatment, as previously discussed in Section 3.3 (hydration properties) [17,31].

### 3.5. Thermal Properties

The thermal properties of all flours are presented in Table 3. Within the temperature range tested, all samples exhibited two peaks, which was similar to that reported by Abebe and Ronda [51] for other tef flours. The first peak (at lower temperatures) corresponds to starch gelatinization (first scan)/retrogradation (second scan after 7 days of storage), and the second one (the smaller peak), which appeared at higher temperatures (90–115 °C), corresponds to amylose–lipid complex dissociation. The Onset (T_O_-gel), Peak (T_P_-gel), and Endset (T_E_-gel) gelatinization temperatures of untreated tef flours were 61.9 °C, 68.9 °C, and 76.7 °C for WTF and 64.51 °C, 71.06 °C, and 78.7 °C for BTF, respectively. As shown in Table 3, the MWT significantly increased each of the thermal parameters (except for T_O_-gel from BTF at 15% MC, that resulted unaltered), as the MC increased. The increase in these parameters has already been reported for other starch sources (potato, cassava, true yam, pea, and lentil) treated by HMT [31], which was attributed to the enhancement of the interactions among the starch chains in amorphous regions (amylose–amylose, amylose–amylopectin, and amylose–lipid) [52]. Villanueva, et al. [17] also suggested that the higher gelatinization temperature of the flours after the MWT may indicate an association and a more stable configuration in a granular structure, which reduce the starch chains’ mobility in the amorphous region. Moreover, the MWT melted the weak crystallites and formed stronger crystallites [53], and as a result, the microwaved samples require greater gelatinization temperatures to disrupt the crystalline regions, leading to an increase in T_O_, T_P_, and T_E_ [31]. The increase of gelatinization temperature observed in moistened treated tef flours would be in consistence with the increase in pasting properties (Section 3.4). Compared to both untreated tef flours, starch gelatinization enthalpy (ΔHgel) did not show any statistically significant difference in microwaved flours. This allows concluding that the differences in hydration and pasting properties found in microwaved samples, even those treated under the strongest conditions (higher MC), cannot be related to a partial gelatinization of the sample during treatment, as sometimes has been reported [14,17,31]. The gelatinization temperature range (ΔT) increased significantly in both treated tef samples due to the most marked increase in T_E_-gel than in T_O_-gel as a result of the treatment. This means the treatment more successfully stabilizes the more perfect and stable crystalline structures than the smaller, more imperfect crystals. These results are in agreement with those reported by Sharanagat, et al. [50] in microwaved sorghum samples. The enthalpy determined for the melting of the amylose–lipid complex (ΔHam-lip) in the first scan was not significantly affected by the MWT and the same happened to the T_P_-am-lip parameter.

A second scan was performed to study the retrogradation enthalpy of amylopectin. The samples were stored for 7 days at 4 ± 2 °C to allow amylopectin retrogradation. The said scan also resulted in two peaks. The main peak corresponded to melting of recrystallized amylopectin (ΔHret), and the second one to the reversible amylose–lipid complex dissociation peak (ΔHam-lip-ret) [54]. No significant differences were observed in ΔHret among all studied samples (4.3–5.0 J/g dm). This indicates that the treatment employed here did not change the ability of amylopectin to reassociate after gelatinization [14]. The values determined for ΔHam-lip in the second scan showed no significant differences among the WTF flours and a slight decrease among the BTF treated at higher MC. ΔHam-lip presented higher values in the second scan (retrogradation) than those determined in the first one (gelatinization), which may be attributed to the better conditions for complex formation after first heating following amylose leaking from starch granules that can occur at temperatures above the gelatinization temperature range [55]. The observations of multiple endothermic peaks and delayed gelatinization temperature of the treated tef flours in both studied scans could be further evidence of the phenomenon of starch molecule reassociation induced by MWT [43].

### 3.6. Fourier Transform Infrared Spectroscopy (FTIR)

The FTIR spectra was used to evaluate the ordered structure of starch and changes to the protein secondary structure. Changes in carbohydrates are observed in the 1100–900 cm^−1^ region and within this region, the bands at 1047 cm^−1^, 1022 cm^−1^, and 995 cm^−1^ have been associated to the starch crystalline structure, the starch amorphous structures, and C–OH bending vibrations (particularly sensitive to water content in starch), respectively [56]. Thus, the band ratios at 1047/1022 cm^−1^ and 1022/995 cm^−1^ have been frequently adopted to measure the proportion of amorphous to ordered molecular starch structure [54]. Table 4 presents the 1047:1022 and 1022:995 ratios for treated and untreated tef flours. Results showed that 1047/1022 cm^−1^ ratio was significantly reduced by MWT, which suggests a lower relative crystallinity of starch in the treated samples. Similar findings were reported by Khuntia, et al. [34], which observed that the application of MW energy to wheat grains reduced the 1047/1022 cm^−1^ ratio compared to raw wheat, attributing this to damage to the crystalline region of starch by the treatment. 1022/995 cm^−1^ ratio remained almost unchanged for the treated samples, and only WTF-25 and BTF-20 (values of 0.932 and 0.918, respectively) were significantly different from their controls (0.911 and 0.901). This increase is related to a higher proportion of amorphous to ordered structure zones in the starch granules [57], and could be attributed to the increase in the proportion of short-chain amylose [56]. In general, lower values for two studied ratios were observed in brown tef samples compared to the white ones.

The changes in protein secondary structure induced by MWT were evaluated in individual bands identified in the 1700–1600 cm^−1^ range (Figure 3), corresponding to amide I band. This band has been widely used to study protein folding, unfolding, and aggregation with infrared spectroscopy due to its intense protein signal and less influence of the side chains [58]. The relative area of each individual conformation is presented in Table 4 and was used to quantify changes due to MWT. The results indicated that modifications on protein secondary structures were more affected by the MC (*p* < 0.001) during the treatment (except β-turn) than by the tef ecotype or the interaction between MC and the used tef flour. The predominant structures in both flours are β-sheets and the random coil and α-helix structures, representing about 80% of the total secondary structure.

A similar trend was observed for both tef ecotypes. The untreated samples had the highest proportion of low-frequency β-sheet (LF-β-sheet) (38.1% and 38.8% for white and brown tef, respectively) and the lowest content of random coil and α-helix and high-frequency β-sheet (HF-β-sheet). A significant decrease of LF-β-sheet content was observed in all MW-treated samples with respect to the untreated flours. This caused a significant increase in random coil and α-helix structures (up to 20% depending on the treatment condition). The said decrease in the ordered structures (β-sheet) and increase in the disordered structure (random coil) may be attributed to the higher degree of protein unfolding and structural flexibility as a consequence of the high water content of the samples and the temperature reached during the treatments [34].

The reduction in LF-β-sheet content was observed in the following order: samples treated at 15% MC < at 20% MC < at 25% MC < Untreated flours. The random structure and α-helix increased in the following order: Untreated flours > at 25% MC > at 20% MC > at 15% MC, similarly for both tef ecotypes. Sun, et al. [59] and Solaesa, et al. [12] also reported an equivalent reduction of β-sheet with a consequent increase of random coil and α-helix for pigeon pea and rice flour treated by MW, respectively.

No significant differences were observed in the β-turn fractions among the studied samples, which suggest that this structure does not seem to depend on experimental conditions applied in this study. From data presented in Table 4 and the spectra shown in Figure 3, it can be seen that the MWT significantly increased the HF-β-sheet content (97% for WTF and 110% for BTF). Moreover, in the 1690–1700 cm^−1^ range, two peaks in all treated samples were observed (except for BTF at 25% MC). The relocations of the HF-β-sheets at this band suggest that there was formation of protein aggregates by MWT [60].

### 3.7. Nuclear Magnetic Resonance (NMR)

The chemical structures of tef flours were examined by NMR spectroscopy, which is a powerful and reliable technique to determine the degree of branching (DB) of the molecular structure of starch in the samples. As presented in Figure 4, the anomeric signals α−(1,6) and α−(1,4) of the glycosidic bonds were clearly visible at 4.80 and 5.12 ppm, respectively. This finding are in accordance with those reported by Xu, et al. [61] and Acevedo, et al. [27] for maize and cowpea starches, respectively. The DB of the untreated WTF and BTF samples were 5.2% and 4.9%, respectively, which are in agreement with data reported for DB that was found to vary from 1% to 5% depending on the botanical source and/or amylose/amylopectin ratio [62]. No significant differences were observed for DB between untreated and samples treated at different conditions, in spite a decreasing trend with the MC of flour during the treatment was observed. This suggests that some glycosidic bonds could be damaged during the MWT, although not to a large extent. Further assays are required to evaluate the effect of MWT on starch molecular structure.

### 3.8. Rheological Properties of the Gels

The rheological properties of the gels formed with the studied flours were deter-mined by dynamic oscillatory tests. Table 5 shows the coefficients G_1_′, G_1_″, and (tan δ)_1_ and the exponents *a*, *b*, and *c*, obtained from fitting the power law model to experimental data obtained from frequency sweeps. Table 5 also includes the maximum stress (τ_max_) within the linear viscoelastic region (LVR) and the stress at the crossover point (G′ = G″, tan δ = 1) obtained from the strain sweeps. The behaviors presented by the studied flours are illustrated in Appendix A. The viscoelastic properties of gels samples were significantly affected by the tef ecotype, MC, and by the interaction between the MC and tef ecotype (see Table 5). The elastic (G_1_′) and viscous (G_1_″) moduli of gels made with WTF were always higher than those made with BTF, denoting a stronger consistency of the former; however, MWT improved significantly (*p* < 0.001) the viscoelastic properties of gels made with tef of the two ecotypes, which reached, in some cases, similar values in both ecotypes, depending on the MC of the flour during the treatment (as consequence of the significant interaction MC x Ecotype). This confirms that the flour treatment has a structuring effect on the gels and allows the modulation of their consistency by controlling the MC during the treatment. The increase in MC consistently showed a positive effect on G_1_′ and G″ values (except for BTF-25). The highest increases in G_1_′ and G_1_″ of WTF (+29% and +16%, respectively) and BTF (+34%, +25%, respectively) were observed in samples treated at 25% and 20% MC, respectively. The increase in the G_1_′ of the gels could be a result of reduced rigidity loss in the swollen starch granule during the MWT, which would tend to enhance interactions between the granules and released amylose [63]. On the other hand, the increase in the loss modulus (G_1_″) could be related to the increased stability of granular integrity caused by the strengthening of the bonds in the swollen granules [64]. The different behavior of BTF-25 was probably due to the partial collapse of the structure of the granules [31].

As shown in Table 5, as MC was increased, the G_1_′ values showed a faster increase than the G_1_″ values, explaining the decrease observed in the loss tangent (G_1_″/G_1_′) values measured from all MW-treated samples (except for BTF-25) with regard to their respective control samples. (tan δ)_1_ < 1 indicates a gel with predominantly elastic over viscous properties, which means that the MWT yielded gels with more pronounced solid-like behavior. These findings place MW-treated tef flours as a promising ingredient for the formulation of GF products, even for breads, as they can increase the elastic behavior of bread dough, with the concomitant benefits on their gas retention capacity, necessary to obtain a well-developed bread with high volume and soft texture. Similar G_1_′, G_1_″, and (tan δ)_1_ values trends were observed for other flours treated by HMT [14,65,66,67].

The exponents “*a*” and “*b*” decreased with increasing MC in both tef flours (Table 5), with “*a*” values always markedly lower than “*b*” values. This indicates that G_1_″ increased at a much higher rate than G_1_′, resulting in a significant increase in the loss tangent (positive “*c*” values) with the angular frequency.

The gels formed from WTF showed higher τ_max_ and crossover values than gels produced from BTF. This means higher stability against shear, as they need a higher stress to destroy their structure and achieve a predominantly viscous behavior. Moreover, as can be seen in Table 5, the results showed a significant increase in τ_max_ and crossover values from samples treated with 15% MC and a significant decrease in both parameters from both tef flours treated with higher water content (20% and 25% MC). This indicates that MWT at 15% MC improved the resistance of the gels to disruption. A similar behavior was observed by Solaesa, et al. [14], who reported that microwaved rice samples with low MC presented higher values of τ_max_ and crossover with respect to the untreated flours. These findings suggest that MWT performed in flours with less water content allows the obtention of stable and consistent gel structures. However, samples treated with higher moisture contents (20% and 25%) showed an opposite behavior. The gels made from these samples presented lower values of τ_max_ and stress at the crossover point, with respect to their control samples. This could be explained due to the partial collapse of the starch granule structure which resulted in less rigid gel [31], similar to the results reported by Vicente et al. [15] and Solaesa et al. [14].

## 4. Conclusions

The treatment conditions employed in this study allowed the modulation of the physical, functional, and microstructural properties of white and brown tef flours. Depending on the MC of the samples during the treatments, some properties were positively and others negatively affected, which allows obtaining a tef flour with a wide range of applications.

A positive effect of increasing MC during the MWT was observed for WAC, WAI, WSI, and SP properties in all treated samples, indicating that flours treated with high MC could favor dough consistency and structure. On the other hand, the pasting properties (PV, BV, SV, and FV) were decreased as the MC increased, with the most significant reductions observed in the samples treated at 25% MC. Moreover, at this MC level the highest delay in PT was observed. The treated flours produced gels with higher G_1_′ and G_1_″ moduli and lower (tan δ)_1_, which indicate a more elastic behavior of these gels, thus favoring the use of this cereal in bread-making. The MWT also led to an increase in gelatinization temperatures from all studied samples, with brown tef being the most affected. FTIR also showed significant differences between microwaved and no-microwaved samples. The treatment conditions employed in this study significantly affected the 1047/1022 cm^−1^ ratio in all samples, while 1022/995 cm^−1^ ratio was found to be affected only in samples treated at 20% and 25% MC. The MWT also modified the secondary structure of proteins in amide I, with the most significant changes observed to samples treated at 15% MC. Future works will be necessary to evaluate the ability of the treated flours to improve the technological, nutritional, and sensory quality of food products.

## Figures and Tables

**Figure 1 foods-12-01345-f001:**
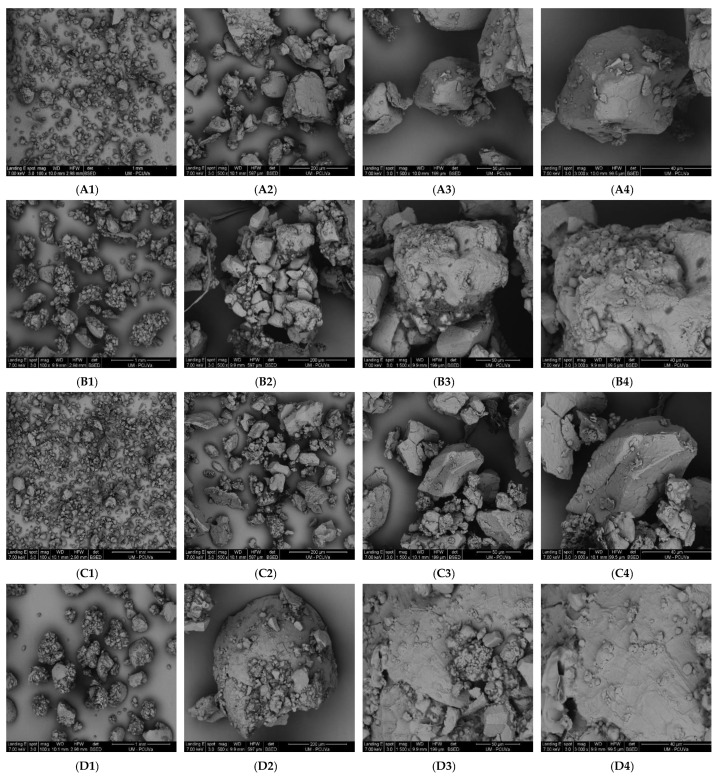
SEM images of selected samples. (**A**) Untreated white tef flour sample (WTF), (**B**) WTF-25 sample, treated at 25% MC, (**C**) Untreated brown tef flour sample (BTF), and (**D**) BTF-25 sample, treated at 25% MC. The numbers 1, 2, 3, and 4 in each image refer to magnifications of 100×, 500×, 1500×, and 3000×, respectively.

**Figure 2 foods-12-01345-f002:**
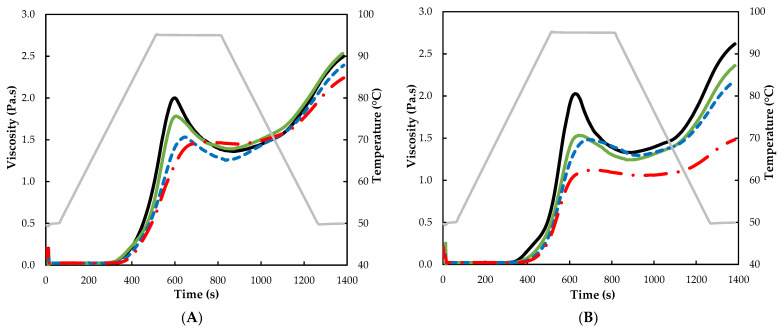
Pasting profiles of untreated and MW-treated tef flours at different moisture content (15%, 20%, and 25% MC). (**A**) White tef flours (WTF) and (**B**) brown tef flours (BTF). Black lines correspond to untreated samples, green lines to samples treated at 15% MC, blue discontinuous lines to samples treated at 20% MC, red discontinuous lines to samples treated at 25% MC, and gray lines to the temperature profile.

**Figure 3 foods-12-01345-f003:**
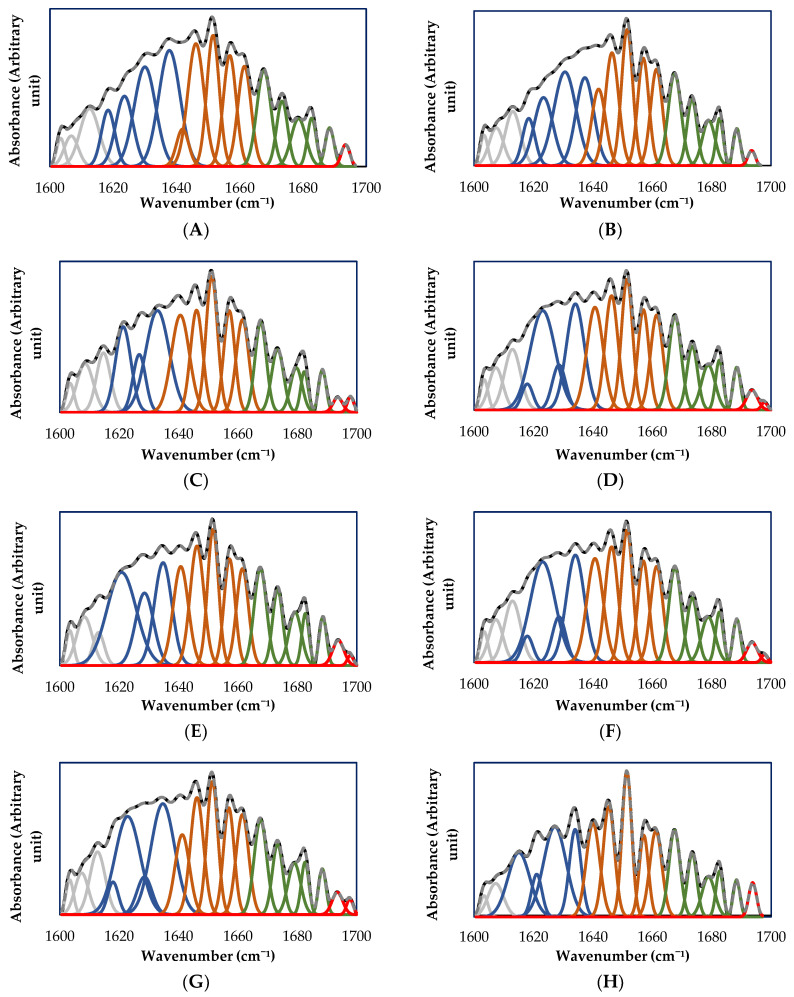
Deconvolved amide I bands of untreated and MW-treated samples. (**A**) Untreated white tef flour (WTF); (**B**) untreated brown tef flour (BTF); (**C**) WTF treated at 15% MC; (**D**) BTF treated at 15% MC; (**E**) WTF at 20% MC; (**F**) BTF at 20% MC; (**G**) WTF at 25% MC; (**H**) BTF at 25% MC. Black lines correspond to deconvolved FTIR spectra; discontinuous gray line to the fitted curves; blue line to β-sheet low frequency (1615–1640 cm^−1^); brown line to random coil and α-helix (1640–1665 cm^−1^); green line to β-turns (1665–1690 cm^−1^); and red line to β-sheet high frequency (1690–1700 cm^−1^).

**Figure 4 foods-12-01345-f004:**
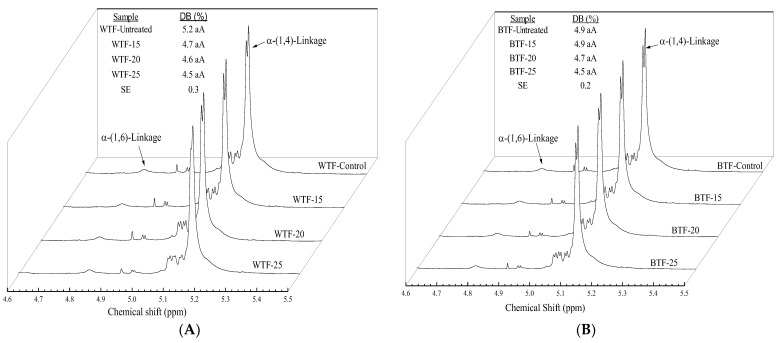
^1^H NMR spectra and the degree of branching (DB %) of untreated and treated flours at different moisture content (15%, 20%, and 25%). (**A**) White tef flours (WTF), and (**B**) brown tef flours (BTF). Different letters indicate statistically significant differences between means at *p* < 0.05. Lowercase letters compare the effect of MC and capital letters compare the effect of tef ecotype.

**Table 1 foods-12-01345-t001:** Effect of microwave treatment on particle size distribution, damaged starch content, amylose content (AC), and hydration properties of tef flours.

Samples	D_50_ (μm)	(D_90_ − D_10_)/D_50_	Damaged Starch (g/100 g)	AC (g/100 g)	WAC (g/g)	WAI (g/g)	WSI (g/100 g)	SP (g/g)
WTF-Untreated	158 aA	2.03 dA	2.28 cA	17.5 aA	1.04 aA	5.50 aA	5.08 aB	5.79 aA
WTF-15	198 bB	1.70 cA	1.80 aB	19.4 bA	1.17 bA	6.50 bA	5.40 bA	6.87 bA
WTF-20	205 cB	1.61 bA	1.99 bB	19.7 bA	1.61 cA	7.63 cA	8.66 dB	8.15 cA
WTF-25	229 dB	1.44 aA	1.93 bB	21.3 cA	1.76 dB	7.57 cB	7.79 cA	8.19 cB
SE	2	0.02	0.03	0.2	0.01	0.06	0.08	0.06
BTF- Untreated	151 aA	2.22 cB	2.30 cA	18.7 aA	1.07 aB	5.83 aB	4.46 aA	6.10 aB
BTF-15	170 bA	1.96 bB	1.54 aA	23.2 bB	1.37 bB	7.59 cB	5.35 bA	8.01 cB
BTF-20	176 bA	1.85 aB	1.56 aA	24.2 bcB	1.58 cA	7.58 cA	6.91 cA	8.16 cA
BTF-25	168 bA	1.96 bB	1.80 bA	25.4 cB	1.71 dA	6.19 bA	11.90 dB	7.03 bA
SE	3	0.03	0.04	0.4	0.01	0.04	0.2	0.05
Analysis of variance and significance (*p*-values)								
F1	***	***	ns	**	ns	ns	ns	ns
F2	**	**	***	*	***	***	***	***
F1 × F2	***	***	***	**	***	***	***	***

WTF: White tef flour; BTF: Brown tef flour. The numbers 15, 20, and 25 in samples column refer to the moisture content (MC) of the samples during the treatment. D50: median diameter; (D90 − D10)/D50: size dispersion. AC (g/100 g): g of amylose content/100 g of starch. WAC: water absorption capacity; WAI: water absorption index; WSI: water solubility index; SP: swelling power. Starch damage, amylose content, WAC, WAI, WSI, SP are referred to flour dry matter. SE: pooled standard error from analysis of variance and significance (ANOVA). Different letters in the corresponding column within each studied factor indicate statistically significant differences between means at *p* < 0.05. Lowercase letters compare the effect of MC and capital letters compare the effect of tef ecotype. ANOVA: *** *p* < 0.001. ** *p* < 0.01. * *p* < 0.05. ns: not significant. F1: tef ecotype factor; F2: MC factor. F1 × F2: Interaction between F1 and F2 factors.

**Table 2 foods-12-01345-t002:** Pasting properties of the studied flours.

Samples	PT	PV	TV	BV	FV	SV
	(°C)	(Pa·s)	(Pa·s)	(Pa·s)	(Pa·s)	(Pa·s)
WTF-Untreated	76.43 aA	2.004 dA	1.352 bA	0.652 dA	2.50 cA	1.15 bA
WTF-15	76.38 aA	1.793 cB	1.399 cB	0.395 cB	2.53 cB	1.13 bA
WTF-20	78.77 bA	1.524 bA	1.252 aA	0.272 bB	2.37 bB	1.12 bB
WTF-25	80.63 cA	1.476 aB	1.448 dB	0.028 aA	2.24 aB	0.79 aB
SE	0.04	0.007	0.006	0.007	0.01	0.01
BTF-Untreated	79.01 aB	2.030 dB	1.329 dA	0.701 dA	2.63 dB	1.30 dB
BTF-15	81.46 bB	1.537 cA	1.242 bA	0.296 cA	2.37 cA	1.13 cA
BTF-20	83.77 cB	1.467 bA	1.290 cB	0.177 bA	2.16 bA	0.87 bA
BTF-25	85.15 dB	1.120 aA	1.053 aA	0.067 aB	1.48 aA	0.43 aA
SE	0.05	0.008	0.003	0.006	0.01	0.01
Analysis of variance and significance (*p*-values)						
F1	**	ns	**	ns	ns	ns
F2	ns	***	ns	***	**	***
F1 × F2	***	***	***	***	***	***

WTF: White tef flour; BTF: Brown tef flour. The numbers 15, 20, and 25 in samples column refer to the moisture content (MC) of the samples during the treatment. PT = Pasting Temperature. PV = Peak Viscosity. TV = Trough Viscosity. BV = Breakdown Viscosity. FV = Final Viscosity. SV = Setback Viscosity. SE: Pooled standard error from analysis of variance and significance (ANOVA). The different letters in the corresponding column within each studied factor indicate statistically significant differences between means at *p* < 0.05. Lowercase letters compare the effect of moisture content and capital letters compare the effect of tef ecotype. ANOVA: *** *p* < 0.001. ** *p* < 0.01. * *p* < 0.05. ns: not significant. F1: tef ecotype factor; F2: MC factor. F1 × F2: Interaction between F1 and F2 factors.

**Table 3 foods-12-01345-t003:** Thermal properties of treated flours and the untreated flours.

Samples	First Scan (Gelatinization)								Second Scan (Retrogradation)					
	ΔHgel (J/g)	T_O_-Gel (°C)	T_P_-Gel (°C)	T_E_-Gel (°C)	ΔT (°C)	ΔHam-Lip (J/g)	T_P_-am-Lip (°C)		ΔHret (J/g)	T_O_-Ret (°C)	T_P_-Ret (°C)	T_E_-Ret (°C)	ΔHam-Lip (J/g)	T_P_-am-Lip (°C)
WTF- Untreated	10.6 aA	61.9 aA	68.9 aA	76.7 aA	14.8 aA	1.1 aA	96 aA		4.4 aA	34.4 bA	51.8 aA	61.3 aA	1.3 aA	96.7 bB
WTF-15	10.9 aB	62.4 bA	69.2 aA	77.6 bA	15.3 aA	1.0 aA	95 aA		4.4 aA	37.6 cA	50.7 aA	61.7 aA	1.4 aA	94.8 aA
WTF-20	10.5 aA	63.4 cA	71.0 bA	81.9 cA	18.5 cA	0.8 aA	95 aA		4.5 aA	33.5 aA	50.2 aA	62.4 bA	1.3 aA	94.5 aA
WTF-25	9.9 aA	66.1 dA	72.2 cA	82.6 cA	16.5 bA	0.8 aA	97 aA		4.8 aA	37.6 cB	49.7 aA	62.4 bA	1.0 aA	95.2 aB
SE	0.3	0.1	0.3	0.2	0.2	0.2	1		0.1	0.2	0.6	0.1	0.3	0.3
BTF- Untreated	9.2 aA	64.51 aB	71.06 aB	78.7 aB	14.2 aA	1.0 abA	95.3 aA		4.2 aA	36.6 bB	50.3 aA	61.9 aB	2.2 cB	95.9 aA
BTF-15	9.6 aA	64.51 aB	71.66 bB	81.5 bB	17.0 bB	1.3 bA	94.9 aA		4.3 aA	38.6 cA	50.3 aA	61.8 aA	2.1 bcA	95.6 aA
BTF-20	9.4 aA	65.97 bB	73.07 cB	83.1 cA	17.1 bA	1.0 abA	96.8 aB		4.7 aA	38.4 cB	51.5 aA	62.1 abA	1.7 bA	94.4 aA
BTF-25	9.0 aA	67.20 cB	74.08 dB	84.4 dB	17.2 bA	0.7 aA	97.3 aA		5.0 aA	34.0 aA	49.6 aA	62.6 bA	1.1 aA	95.0 aA
SE	0.2	0.06	0.02	0.3	0.3	0.1	0.8		0.3	0.3	0.7	0.1	0.1	0.8
Analysis of variance and significance (*p*-values)														
F1	***	*	**	ns	ns	ns	ns		ns	ns	ns	ns	*	ns
F2	ns	**	*	***	***	ns	ns		*	ns	ns	***	ns	*
F1 × F2	ns	***	ns	**	**	ns	ns		ns	***	ns	ns	ns	***

WTF: White tef flour; BTF: Brown tef flour. The numbers 15, 20, and 25 in samples column refer to the moisture content (MC) of the samples during the treatment. ΔHgel = Enthalpy of gelatinization. TO–gel, TP–gel, TE–gel: Onset, peak and endset temperatures of gelatinization. ΔT = (TE–gel – TO–gel). ΔHam–lip = Enthalpy of the amylose–lipid dissociation. TP–am–lip = Peak temperature of the amylose–lipid complex dissociation. ΔHret = Enthalpy of melting of retrograded amylopectin. TO–ret, TP–ret, TE–ret: Onset, peak and endset temperatures of melting of retrograded amylopectin. ΔHgel, ΔHret, ΔHam–lip are given in J/g dry matter. SE: Pooled standard error from analysis of variance and significance (ANOVA). The different letters in the corresponding column within each studied factor indicate statistically significant differences between means at *p* < 0.05. Lowercase letters compare the effect of moisture content and capital letters compare the effect of tef ecotype. ANOVA: *** *p* < 0.001. ** *p* < 0.01. * *p* < 0.05. ns: not significant. F1: tef ecotype factor; F2: MC factor. F1 × F2: Interaction between F1 and F2 factors.

**Table 4 foods-12-01345-t004:** Starch bands and secondary structure content of treated and untreated flours from FTIR analysis.

Samples	Starch Bands			Secondary Protein Structures in Amide I Region (cm^−1^) (%)			
	IR 1047/1022	IR 1022/995		LF β-Sheet	Random Structure & α-Helix	β-Turn	HF β-Sheet
WTF- Untreated	0.818 cB	0.911 aA		38.1 cA	40.4 aA	20.5 abA	0.95 aA
WTF-15	0.803 bB	0.921 aB		30.0 aA	47.4 dA	20.7 bA	1.87 bB
WTF-20	0.786 aA	0.916 aA		33.1 bA	45.3 cA	19.7 aA	1.87 bA
WTF-25	0.793 aB	0.932 bB		35.1 bA	43.2 bA	20.1 abA	1.63 bA
SE	0.003	0.003		0.7	0.6	0.3	0.09
BTF- Untreated	0.800 dA	0.901 abA		38.8 cA	40.1 aA	20.2 aA	0.85 aA
BTF-15	0.773 bA	0.892 aA		30.1 aA	49.1 cA	19.5 aA	1.31 bA
BTF-20	0.785 cA	0.918 cA		35.3 bA	43.7 bA	19.4 aA	1.54 bcA
BTF-25	0.764 aA	0.905 bA		36.4 bcB	42.7 abA	19.1 aA	1.79 cA
SE	0.001	0.003		0.9	0.9	0.4	0.08
Analysis of variance and significance (*p*-values)							
F1	**	***		ns	ns	*	ns
F2	**	ns		***	***	ns	***
F1 × F2	***	***		ns	ns	ns	**

WTF: White tef flour; BTF: Brown tef flour. The numbers 15, 20, and 25 in samples column refer to the moisture content (MC) of the samples during the treatment. LF: Low frequency; HF: High frequency. SE: Pooled standard error from analysis of variance and significance (ANOVA). The different letters in the corresponding column within each studied factor indicate statistically significant differences between means at *p* < 0.05. Lowercase letters compare the effect of moisture content and capital letters compare the effect of tef ecotype. ANOVA: *** *p* < 0.001. ** *p* < 0.01. * *p* < 0.05. ns: not significant. F1: tef ecotype factor; F2: MC factor. F1 × F2: Interaction between F1 and F2 factors.

**Table 5 foods-12-01345-t005:** Rheological properties of the studied flours.

Samples	G_1_′ (Pa)	*a*	G_1′_″ (Pa)	*b*	(tan δ)_1_	*c*	τ_max_ (Pa)	Crossover (Pa)
WTF-Untreated	422 aB	0.019 cA	59.6 aB	0.293 bA	0.141 cA	0.2736 aB	615 cB	657 bB
WTF-15	453 bB	0.017 cA	61.1 bA	0.300 cA	0.135 bA	0.2799 bB	647dB	673 bB
WTF-20	503 cB	0.006 bA	65.0 cA	0.299 cB	0.129 aA	0.2924 cB	546 aB	572 aB
WTF-25	546 dB	−0.014 aA	69.2 dB	0.287 aB	0.127 aA	0.3008 dB	578 bB	592 aB
SE	3	0.002	0.3	0.001	0.001	0.0007	4	7
BTF-Untreated	332 aA	0.089 dB	51.5 aA	0.305 dB	0.155 bB	0.2161 cA	222 cA	256 cA
BTF-15	420 cA	0.062 aB	60.3 bA	0.298 cA	0.144 aB	0.2367 dA	283 dA	331 dA
BTF-20	445 dA	0.069 bB	64.3 cA	0.281 bA	0.145 aB	0.2119 bA	207 bA	230 bA
BTF-25	371 bA	0.078 cB	60.6 bA	0.276 aA	0.164 cB	0.1978 aA	90 aA	121 aA
SE	2	0.001	1	0.001	0.001	0.0008	1	6
Analysis of variance and significance (*p*-values)								
F1	**	***	ns	ns	***	***	***	***
F2	ns	ns	**	*	ns	ns	ns	ns
F1 × F2	***	***	***	***	***	***	***	**

WTF: White tef flour; BTF: Brown tef flour. The numbers 15, 20, and 25 in samples column refer to the moisture content (MC) of the samples during the treatment. G_1_′ (elastic modulus), G_1_″ (viscous modulus), and (tan δ)_1_ (loss tangent) are the coefficients obtained from fitting the frequency sweeps data to the power law model and represent the moduli and loss tangent values at a frequency of 1 Hz. The *a*, *b*, and *c* exponents quantify the dependence degree of dynamic moduli and the loss tangent with the oscillation frequency. τ_max_ represents the maximum stress tolerated by the sample in the LVR. SE: Pooled standard error from analysis of variance and significance (ANOVA). The different letters in the corresponding column within each studied factor indicate statistically significant differences between means at *p* < 0.05. Lowercase letters compare the effect of moisture content and capital letters compare the effect of tef ecotype. ANOVA: *** *p* < 0.001. ** *p* < 0.01. * *p* < 0.05. ns: not significant. F1: tef ecotype factor; F2: MC factor. F1 × F2: Interaction between F1 and F2 factors.

## Data Availability

The data presented in this study are available on request from the corresponding author.

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
