# Peer review of "Changes on Techno-Functional, Thermal, Rheological, and Microstructural Properties of Tef Flours Induced by Microwave Radiation—Development of New Improved Gluten-Free Ingredients"

_foods, 2023, doi:10.3390/foods12061345_

Round 1
Reviewer 1 Report
Manuscript tile: Changes on techno-functional, thermal, rheological, and microstructural properties of tef flours induced by microwave radiation. Development of new improved gluten-free ingredients
The above-titled manuscript deals with techno-functional, thermal, rheological, and microstructural properties of tef flours induced by microwave radiation with a recommendation as novel gluten-free flour. The study is well presented with interesting findings. However, authors must consider below suggestions to improve the quality of the manuscript.
Title can be revised as Changes on techno-functional, thermal, rheological, and microstructural properties of tef flours induced by microwave radiation: Development of new improved gluten-free ingredients
Abstract
Authors should provide the conclusions and recommendations for the study in abstract
Introduction
Well, presented with appropriate research gap, and hypothesis
2.3 Particle size distribution: provide the detailed methodology along with appropriate citation
2.5 Damaged starch and apparent amylose content: provide the detailed methodology
2.6 Hydration properties: provide the detailed methodology
2.7 Pasting properties: provide the detailed methodology and temp used
2.11 Rheological properties: provide the detailed methodology
Sections 3.1 and 3.2. are well written
3.4. Pasting properties: discussion on this section must be improved
Other results section is well presented with interesting findings. Authors also compared with the literature
References must be cross-checked and must be adjusted according to the journal guidelines
Author Response
Please, refer to the attached document

Reviewer 2 Report
This manuscript evaluated the changes on techno-functional, thermal, rheological, and micro-structural properties of tef flours induced by microwave radiation for the development of new gluten-free ingredients. The topic is interesting and the manuscript is well-designed. It needs a minor revision.
1. Please provide the scientific name of white and brown tef grains.
2. Section 2.2; Did the authors use distilled water or salt water?
3. L 136, kV, not keV.
4. Please represent the Methods and Results and discussion in the same order.
5. L 238, Please check the definition of annealing.
6. L 240, "said" should be removed.
7. Figure 2 and Table 3 should be placed after section 3.4 and 3.5, respectively.
Author Response
Please, refer to the attached document
